# Evaluation of 3D Printability and Biocompatibility of Microfluidic Resin for Fabrication of Solid Microneedles

**DOI:** 10.3390/mi13091368

**Published:** 2022-08-23

**Authors:** Atabak Ghanizadeh Tabriz, Beatriz Viegas, Michael Okereke, Md Jasim Uddin, Elena Arribas Lopez, Nazanin Zand, Medhavi Ranatunga, Giulia Getti, Dennis Douroumis

**Affiliations:** 1Faculty of Engineering and Science, School of Science, University of Greenwich, Chatham Maritime, Chatham ME4 4TB, UK; 2CIPER Centre for Innovation and Process Engineering Research, Kent ME4 4TB, UK; 3School of Science and Technology, NOVA University Lisbon, 2829-516 Almada, Portugal

**Keywords:** 3D printing, Digital Light Processing, microneedles, piercing, mechanical properties, biocompatibility

## Abstract

In this study, we have employed Digital Light Processing (DLP) printing technology for the fabrication of solid microneedle (MN) arrays. Several arrays with various geometries, such as cones, three-sided pyramids and four-sided pyramids, with different height to aspect ratios of 1:1, 2:1 and 3:1, were printed. Post-processing curing optimizations showed that optimal mechanical properties of the photocurable resin were obtained at 40 °C and 60 min. Ex vivo skin studies showed that piercing forces, penetration depth and penetration width were affected by the MN geometry and height to aspect ratio. Cone-shaped MNs required lower applied forces to penetrate skin and showed higher penetration depth with increasing height to aspect ratio, followed by three-sided and four-sided printed arrays. Cytotoxicity studies presented 84% cell viability of human fibroblasts after 2.5 h, suggesting the very good biocompatibility of the photocurable resin. Overall, DLP demonstrated excellent printing capacity and high resolution for a variety of MN designs.

## 1. Introduction

Three-dimensional (3D) printing, usually recognized as additive manufacturing, is a layer-by-layer fabrication process in which different materials, such as metals, polymers and ceramics, can be processed to produce complex 3D objects [1]. Three-dimensional printing has superior fabrication flexibility in comparison to other conventional fabrication methods. Therefore, such a rapid prototyping approach allows the fabrication of complex and precise geometries in a timely manner from computer-aided design (CAD) models [2]. 

Nowadays, several 3D printing techniques, such as vat photopolymerization, for example, Laser Stereolithography (SLA) and/or Digital Light Processing (DLP), are utilized to allow the fabrication of 3D physical objects. SLA cures and solidifies photo-polymeric resins using an ultraviolet (UV) laser and has been recognized as one of the most used photopolymerization 3D printing techniques due to its high precision and accuracy [3]. The technique is controlled by the dimension of the laser focal point, and SLA printers had a limited resolution of 100 μm, restricting the resolution for the fabrication of several structures, until recently, when the laser focal point was reduced to 25–86 μm [4]. Nevertheless, instead of using a laser, a light projector can be implemented to expose the entire layer to UV light simultaneously, and, in this case, the process name changes to Digital Light Processing. This printing method allows higher-resolution prototyping due to comprising very small pixels (range 20 to 100 μm), fast processing and excellent surface finishing and can be used for the fabrication of (bio) medical devices. Subsequently, SLA/DLP methods require a post-curing process after printing to ensure the solidification of any uncured resin [3]. Hence, further studies of print optimization and curing parameters have been developed to evaluate their influence on quality features [4].

Therefore, the vat photopolymerization printing concept has revolutionized several fields, such as pharmaceutical and biomedical, with significant research outcomes [3,5,6]. In the pharmaceutical field, these techniques can be applied in the fabrication process of Transdermal Drug Delivery (TDD) systems. TDD systems can substitute typical intramuscular or subcutaneous administration routes while being patient-friendly. A typical example is MN patches, which are micron-scale needles with powerful delivery capabilities.

The first MN array was introduced in the 1970s and was made of silicone. Since then, MNs have been developed and fabricated with different materials, such as metal, polymer, glass and ceramic [7]. However, the safety and biocompatibility of these materials for biomedical applications are important factors that must be considered. Miniature devices can be fabricated by adjusting their dimensions and shape according to the required specifications; however, the developed MN design should ensure successful and painless skin penetration during the administration of actives [8]. In addition, the MN dimensions must enable precise and efficient tissue localization delivery combined with low manufacturing costs per MN array. 

Furthermore, MN types can be divided according to the various methods of drug delivery as solid, hollow, coated, porous, hydrogel-forming and polymer or dissolvable arrays [9,10,11,12]. Solid MNs are appropriate for penetration and can be used to increase permeability before drug application. This type of MN forms micron-scale pores in the skin surface after being applied and removed. These pores allow the slow diffusion of drugs into the body. Moreover, the choice of the material and geometry of these types of MNs should provide appropriate mechanical strength and reduce their insertion force into the skin by improving tip sharpness. 

As previously stated, MNs’ material composition is one of the most important aspects regarding their application and fabrication process. It should ensure that MNs are biocompatible and present suitable mechanical strength. However, most photosensitive resins have demonstrated poor biocompatibility [13], causing immunological responses after the post-curing process [14]. Moreover, not all of the currently used resins allow the incorporation and release of drug substances after polymerization and, therefore, cannot be used in pharmaceutical applications. Consequently, it is essential to identify a biocompatible printable resin with high resolution, good mechanical properties and good printability [15]. To meet the aforementioned specifications, the development of biocompatible resins could be an option. An example of a biocompatible resin is the Matrigel-coated WaterShed resin utilized by Au et al., to evaluate its cytocompatibility on C2C12 myoblast cells [16]. The Class I acrylic resin and Dental SG have also been successfully used to 3D print MN patches for intradermal insulin delivery using SLA, developed by Economidou et al. [8].

The aim of this work was to evaluate the capacity of DLP using a photocurable microfluidic resin for the printing of microneedle arrays of various geometries and height to aspect ratios and investigate their effects on the mechanical properties and piercing capacity of the fabricated arrays. The cytotoxicity of the resin was also investigated to verify that it is safe for skin penetration and non-toxic to the skin. 

## 2. Materials and Methods

### 2.1. Materials 

The MNs were 3D-printed using microfluidic resin, MiiCraft BV-007A (Young Optics Inc., Hsinchu, Taiwan). Human dermal fibroblasts (HDFs), human dermal fibroblast growth medium and trypsin EDTA solution (1×) were purchased from Sigma Aldrich (Sigma Aldrich, Rochester, UK). The ReadyProbes^®^ Cell Viability Imaging Kit (Blue/Green) was purchased from Fisher Scientific™ (Fisher Scientific, Loughborough UK). The porcine skin was purchased freshly from a local slaughterhouse (Cheale Meats Ltd., Essex, UK).

### 2.2. Computer-Aided Design (CAD)

Three MN geometries were designed using the SolidWorks CAD software (V20 Dassault Systems, Waltham, MA, USA). Based on the needle shapes of each MN patch design, they were named ‘cone’, ‘3-sided pyramid’ and ‘4-sided pyramid’. The MNs’ dimensions are represented in Table 1. Each MN was designed with a height to aspect ratio of 1:1, 2:1 and 3:1.

The MNs were designed on a square patch of 15 mm × 15 mm × 0.5 mm symmetrically, comprising 7 rows and 7 columns, with a total number of 49 needles per design. The interspacing between the needles was adjusted at 1.85 mm. On the other hand, cylinders were also designed and printed, respectively, for compression testing. The cylinder design was in accordance with ASTM D695-15 standards, where the cylinders had a diameter of ø12.7 mm and a length of 25.4 mm

### 2.3. Additive Manufacturing of Microneedles 

For the printing of the MNs and the compression test specimens, a MAX X27 DLP (Asiga, Erfurt, Germany) printer, equipped with an LED wavelength of 385 nm and a vat of uncured microfluidic resin, was used. The printer software Asiga Composer (Asiga, Erfurt, Germany) was applied to slice the models and to generate the printing files. The MNs’ printing orientation was set to 45° on the printing platform with the aid of a support, which was selected based on our previous studies [4]. The slicing/layer thickness was adjusted to 25 µm and the MNs were printed with a UV intensity set to 3 mW/cm^2^ and a layer exposure time of 3.3 s accordingly. The first layer exposure time was set to 112 s with identical UV intensity to ensure the appropriate adherence of the MNs to the printing platform. The generated printing files were then exported to the printer for fabrication. 

Once printed, the MNs were removed from the printing platform gently and immersed in an isopropanol bath for 10 min to wash out any uncured resin residues. The support structures were then gently removed and the MNs were then left to dry in an ambient environment for 60 min. Then, the printed MNs were cured for 60 min and at 40 °C, which were the optimized curing settings obtained, in a UV-A heated chamber (MeccatroniCore BB Cure Dentalstation, GoPrint3D, Ripon, North Yorkshire, UK) with a wavelength of 405 nm. 

### 2.4. Compression Test 

In order to evaluate the curing settings’ effect on the mechanical properties of the microfluidic resin, several solid cylinders were designed, printed and cured at different times and temperatures. The predetermined curing times used were 30 min, 45 min and 60 min and were combined with pre-set chamber temperatures of 40 °C, 50 °C and 60 °C. A total of 45 samples were analyzed, 5 samples for each respective time and temperature. The cylinders were subjected to a compression test using the Tinius Olsen testing machine (Salfords, Surrey, UK), set up with two flat test plates and a 25 kN load cell, following the standard test method for compressive properties of rigid plastic, ASTM D695. 

Each cylindrical specimen was placed between the two plates and a compression force was applied alongside their longitudinal axis at a constant speed of 1.5 mm/min. For each sample, the applied force value and position were recorded and presented by the Horizon software (Horizon Software, Salfords Surrey, UK). The obtained data were analyzed and, therefore, true stress and true strain values were calculated. The compressive modulus, as well as the yield strength of each sample, was calculated according to the 0.5% offset curve method.

### 2.5. Scanning Electron Microscopy (SEM)

Scanning electron microscopy (Hitachi SU8030, Tokyo, Japan) was utilized to investigate the quality of the 3D-printed MNs. MNs were kept secured on an aluminum stub with a conductive carbon adhesive tape (Agar Scientific, Stansted, UK). The MNs were then examined via SEM and images of the MNs were captured by an electron beam accelerating voltage of 1.0 kV and magnification of 80×.

### 2.6. Microneedle Axial Force Mechanical Testing 

To achieve the maximum force value that could be applied to the MNs, fracture tests were carried out. The MNs were subjected to an axial force test using the Tinius Olsen testing machine (Redhill, Surrey, UK), with the same set-up as the compression tests and following the identical standard test method. The MN’s base was glued to the moving plate using double-sided tape and pressed against the static plate until the needles fractured, at a constant speed of 1.5 mm/min. A total of 5 samples, cured with the optimized settings, were analyzed, and for each one, the applied force value and position were recorded and presented by the Horizon software. 

### 2.7. Preparation of Porcine Skin Samples

The skin samples were collected freshly from a local slaughterhouse (Forge Farm Ltd., Kent, UK) and any hair removed using a razor blade. For the skin piercing test, full-thickness skins (1 mm) were excised using a Dermatome (Padgett Dermatome, Integra Life^TM^ Sciences Corporation, Princeton, NJ, USA). After removing a thick fatty layer of tissue, full thickness was then transferred in a buffer saline of pH 7.4 to the texture analyzer to perform the piercing test. 

### 2.8. Ex Vivo Piercing of Porcine Skin

The piercing capacity of 3D-printed cone-shaped, 3-sided pyramid and 4-sided pyramid MNs was investigated using a texture analyzer, the HD plus Texture Analyser (Stable Micro Systems, Surrey, UK). The compression type of application force included the pre-test, test speed and post-test speeds of 1.00 mm/s, 0.1 mm/s, 1 mm/s, respectively. Double-sided adhesive tape was used to fix the MNs onto the probe, while full-thickness skins were placed onto the petri dishes. Each experiment was repeated 6 times. 

### 2.9. Optical Coherence Tomography (OCT)

OCT (Carl Zeiss India Pvt. Limited, Bangalore, India) was utilized to investigate the penetration capability of the 3D-printed MNs into the freshly excised mouse (2–3 weeks old) dorsal skin. Skin hair was removed using an electric razor (Panasonic, Bracknell, UK) in order to avoid any penetration issues of the laser beam into the deeper tissue parts (2 mm). Then, 3D-printed MNs were inserted into the skin tissue manually. The penetration depths and widths of each dimension of the 3D-printed MNs were determined using a microscope attached to the OCT instrument. False color was used to distinguish the penetrating layers (e.g., stratum corneum, epidermis, dermis, etc.) and 3D-printed MNs using Adobe Photoshop CS6^®^ software (Adobe Systems Inc., San Jose, CA, USA). A single 3D-printed MN array (7 × 7) was used for the OCT analysis. 

### 2.10. Cytotoxicity Studies and Fluorescence Cell Imaging

To evaluate the cytotoxicity of the MNs, circular disks were printed using the respective resin. The disks were designed with a diameter of 6 mm and 0.6 mm thickness in order to fit in a 96-well plate. Each side of the disks was sterilized under UV light for 1 h and subsequently soaked in fibroblast growth medium for 48 h before placing them into a 96-well plate. Human dermal fibroblast cells were harvested from the main culture at low passage (<10) at confluency of 80–90% and >90% viability. They were suspended in fresh media and 100 mL total volume with a concentration of 1 × 10^5^ cells/well was added to each well containing disks. The cells were then incubated for 2.5 h and stained with blue and green stain (Thermo Fisher Scientific, Cat# R37609) according to manufacturer guidelines. Briefly, 10 mL of a mixture containing an equal number of drops of each dye per milliliter was added to each well. The plate was then incubated at 37 °C and 5% CO_2_ for 30 min before taking images using a ZOE Fluorescent Cell Imager (Bio-Rad Laboratories, Watford, UK) at 4× magnification. Cells colored in blue represented the total amount of cells while those in green the damaged cells. The experiments were performed in triplicate. Cell viability was calculated as follows: Cell viability (%) = [Blue cells/(Blue cells + Green cells)] × 100

## 3. Results

### 3.1. Additive Manufacturing of Microneedles 

In the current study, DLP was selected as the printing technology due to the higher-resolution prototyping, fast processing and excellent surface finishing. For the optimization of the printing process, the applied UV laser (385 nm) intensity was set at 3 mW/cm^2^ and the curing of resin layers took place at an exposure time of 3.3 s. The layer thickness identified as Z axis resolution (slice thickness) was defined as 25 µm. The X-Y resolution of the printing refers to the pixel size of the DLP projector at the exposure location of the UV light and the resin. Compared to our pervious study, the Z and X-Y resolutions were enhanced from 50 and 50 µm to 25 and 27 µm, respectively. Such changes in resolution resulted in the fabrication of finer structures and, most importantly, more defined and sharper tips for the MNs. As shown in Figure 1, three MN designs were created, namely cones, three-sided pyramids and four-sided pyramids. In Figure 2, the SEM images illustrate that all 3D-printed MNs comprised very fine features with excellent tip sharpness, which is crucial for skin penetration. The layers of each printed structure appeared very distinct and consistent.

### 3.2. Optimization of Post-Printing Curing Conditions for Enhanced Mechanical Properties 

During the application on the skin, the MN patches are subjected to compressive forces along their longitudinal axis to pierce the skin and, therefore, they must be strong enough to be applied safely and to reduce the chances of breakage. Thus, to prevent failure during application, their mechanical properties need to be evaluated. As mentioned in previous studies [5], the post-printing curing process affects the cross-link density and, hence, some mechanical properties, such as the elastic modulus and ultimate strength. Consequently, a standard compression test was conducted, using cylinder samples, which were printed and cured with different times and temperatures. Subsequently, the MNs’ mechanical properties were assessed to evaluate the optimized curing settings that enhance the mechanical strength of the material.

Throughout the compression testing of the specimens, it was concluded that the maximum load cell (25 kN) of the testing machine was not sufficient to achieve the value of the ultimate strength for some of the specimens, since some of the specimens did not appear to have any fracture in their structure. Due to this reason, the maximum elastic point of the material, the yield strength, was determined. Yield strength was evaluated via the following equation.
$$\mathrm{Yield}\;\mathrm{strength}\;\left(\mathrm{at}\;\mathrm{maximum}\;\mathrm{elastic}\;\mathrm{point}\right)=\frac{\mathrm{Force}}{\mathrm{cross}-\sec\mathrm{tional}\;\mathrm{area}}$$

The findings suggested that the exposure time has a significant influence on the properties of the resin, rather than the curing chamber temperature. Therefore, increasing the exposure time can enhance the mechanical properties since it increases the degree of cross-linking. Furthermore, the influence of chamber temperature becomes weaker with increasing exposure time. This is possible to conclude since the yield strength values for the lower and higher temperature, obtained for 60 min, have a smaller gap compared with other values. As shown in Figure 3, the maximum values obtained for both compressive modulus and yield strength were 1.34 ± 0.02 GPa and 45.69 ± 0.25 MPa, respectively, which corresponds to curing at 40°C and 60 min each. These post-curing settings were considered to be the most suitable for curing in order to achieve the best mechanical performance for the particular resin. As expected, the mechanical properties of MNs fabricated by SLA were higher compared to those reported for polymeric arrays such as hyaluronic acid [14].

According to the obtained results, the post-printing curing process influences the mechanical properties of the resins and thus a systematic evaluation is required. This step played a significant role in the photopolymerization of the printed MNs since it enhanced their mechanical properties, which lowered the possibilities of failure during application.

### 3.3. Piercing Studies in Porcine Skin

All MN arrays could successfully pierce the porcine skin without being fractured or broken during the testing. This was attributed to the production of microchannels without damaging the geometry of the needles [17,18]. However, the piercing force slope was constantly changing before the insertion of the MN tips due to increased skin resistance until a sharp drop in load occurred, which indicated skin penetration. The findings were in accordance with a previously reported study where MN arrays with similar designs were used [4].

As shown in Figure 4a–c, for all 3D-printed MN arrays, the piercing force varied based on the height to aspect ratio (HAR) and MN design. The piercing curves appeared different for each design but similar for identical designs with varying HARs. In Figure 4d, it can be clearly seen that the HAR plays a significant role, irrespective of the printed design. The lower the HAR, the less the required force for skin penetration, and hence the descending force order was 1:1 > 2:1 > 3:1.

Furthermore, the penetration force was affected by the design, where the cones required lesser forces to penetrate the skin in comparison to three-sided pyramids. The four-sided pyramids required relatively higher forces. Nevertheless, for all the designs, the penetration forces varied from 30 to 43 mN per needle of each array. Figure 5 shows photos of all MN arrays with noticeable skin pores at the end of the piercing studies, which confirmed their capacity to penetrate the skin.

### 3.4. OCT Studies

Further evaluation of the piercing capacity of 3D-printed MNs involved skin penetration studies via OCT. It is known that OCT is considered to be a non-destructive and non-invasive method that can capture high-resolution images from the deeper tissues [19,20]. It has been previously employed to study the skin penetration behavior of polymeric [21,22] but also 3D-printed MN arrays [23].

As can be seen in Figure 6, Figure 7 and Figure 8, all three designs (i.e., cone, three-sided pyramid and four-sided pyramid) of 3D-printed MNs showed reproducible and uniform piercing with distinct holes into the skin. Though a manual application force was applied, all shapes presented penetration depths varying within 680.1 ± 33.9 and 833.2 ± 24.8 µm. The variations in the penetration depth were attributed to the efficient penetrating capacity of the different MN designs and the creation of microchannels into the skin tissue.

Table 2 shows the penetration capacity of different 3D-printed MNs in mouse skin, which varied from 68 to 83% (based on the total MN length). As expected, it was observed that MN designs (e.g., base length) and HAR play an important role in the penetration of the skin. The higher the HAR, the higher the penetration capacity and thus the penetration depth. For example, for cone-shaped MNs, the penetration depth was 76.0% and 83.3% for the 1:1 and 3:1 ratios, respectively.

The penetration capacity varied according to the MN designs, where cone shapes showed better skin penetration than three- and four-sided pyramid shapes, respectively. Similar findings were obtained for the penetration width, which increased with increasing HAR. However, the penetration width was higher for four-sided compared to three-sided and cone geometries. This was attributed to the higher volume of the four-sided MNs in comparison to the other shapes, which results in higher volumetric expansion of the skin [24]. The findings in Table 2 are of value as the combined effect of the MN geometry and HAR should be considered when delivering active molecules to the dermis [25]. High penetration depths combined with higher penetration widths can be used to administrate larger drug amounts to the skin. This is because previous studies demonstrated that skin behaves as a deformable porous medium and the volumetric expansion corresponds to the amount that can be administrated [24]. Nonetheless, OCT was proven to be a valuable to tool to evaluate and obtain a better understanding of the effect of the printed geometries in skin penetration.

The piercing capacity of 3D-printed MN designs in this work appears to be far better when compared to similar studies for dissolvable MN patches. For example, Lau et al. (2017) developed multi-layered pyramidal dissolvable patches with a total length of 500 μm, but the penetration capacity was limited to 30%, with only 150 μm depth in animal skin [26]. In another study, chitosan MN patches fabricated by Chen et al. (2012) displayed only 200 μm with a piercing capacity of 25% (total MN length was 600 μm) [27]. It is apparent that DLP-printed microneedles provide not only better mechanical strength but also superior piercing to polymeric dissolvable MNs. This is an important detail that should be taken into account when designing MN patches for the administration of drug substances as it will affect the drug loading, drug release rates and, most importantly, the pharmacokinetic results.

### 3.5. Microneedle Axial Force Mechanical Testing

In order to assess the security and confidence level of the MN patches, the fracture force was evaluated to estimate the failure probability during the course of application. All MN designs were tested to compare their mechanical behavior and evaluate the effect of needle geometry and the HAR.

The data recorded from the axial force measurements are displayed in Figure 9. The fracture point of the printed MNs was considered as the evident variation in the slope in the presented curve. For all the designs, the slope is straight until it reaches the failure point. This means that the needles are mechanically stable, without any evident plastic deformation, and can resist the applied forces.

A careful look at Figure 9b illustrates that 1:1 HAR presented the highest fracture forces irrespective of the needle design, followed by the HAR of 2:1 and 3:1, respectively. Interestingly, the 2:1 and 3:1 HAR showed a significant drop in the fracture force. Among the different designs, the four-sided pyramid appeared to be the most resistant to fracture, with a decreasing order of four-sided pyramid > cone > three-sided pyramid. As expected, the fracture force decreased as a consequence of a smaller base surface area and therefore the decrease in the base cross-section area to support the tip throughout the axial mechanical test. Previous studies demonstrated lower axial force values even with the same geometry and ratio. An example of this was the conical carboxymethylcellulose (CMC) MNs and polylactic acid (PLA) MNs developed by Demir et al. [28], with forces of fracture of 0.1 N/needle and 0.5 N/needle, respectively, 40 times lower than the value obtained here. Other biodegradable polymers were used to manufacture MNs, such as polylactic-co-glycolic acid (PLGA) and sodium alginate (SA); however, both showed lower failure forces compared to our findings. Therefore, SLA resins offer better mechanical properties compared with conventional polymeric MN arrays [14].

### 3.6. Safety Index

The margin of safety can be a useful index to evaluate the MNs’ safety for application. The index was calculated using the values obtained from previous studies and the following equation:(1)$$Margin\;of\;safety\;=\frac{F_{fracture}}{F_{Insertion}}$$
where $F_{fracture}$ is the maximum force that the MN can support without breaking, and $F_{Insertion}$ is the force needed to pierce the skin. To ensure a safe application, the values obtained for the margin of safety should be higher than one. In addition, higher values reveal more reliable and strong MN systems.

As presented in Table 3, the four-sided pyramid and the cone design with a 1:1 HAR had the highest margin of safety, 427.58 and 426.16, respectively. It is obvious that increasing the HAR makes the MNs weaker and causes a lower index. The obtained values showed a dramatic reduction in the safety Index with the increase in the HAR, where, for example, from 426.16 for the 1:1 HAR cone-shaped MNs, it was reduced to 19.70 for the 3:1 HAR, which is approximately 22 times lower. The lowest value obtained corresponded to the three-sided pyramid with a 3:1 HAR design (14.00), which still is >1, indicating the safety of the MN arrays.

### 3.7. Cytotoxicity Studies

The biocompatibility of MNs is a critical factor that must be considered and investigated prior to any human trials. According to our knowledge, there are no biocompatibility/cytotoxicity studies on the specified resin used in this study. Therefore, the cytotoxicity of the microfluidic resin was investigated using HDFs, where cell viability was evaluated through fluorescence microscopy. As shown in Figure 10 and Figure 11, the seeded cells on sterile microfluidic resin disks showed cell viability of 68 ± 6.01%. However, on the other hand, the disks soaked in culture media prior to cell seeding showed enhanced cell viability of 84 ± 3.55%, which indicates the better surface suitability and biocompatibility of the disks. In comparison to the control, which presented viability of 97 ± 2.61%, there was a drop of 13% in cell viability. However, such viability values have been reported to be acceptable in the literature for MN applications [29]. Overall, the cytotoxicity result on the resin was promising; however, further studies on the specified microfluidic resin may provide more comprehensive data on cytotoxicity.

## 4. Conclusions

Microneedle arrays of various geometries, such as cone, three-sided and four-sided, with different height to aspect ratios varying from 1:1, 2:1 and 3:1, were printed using Digital Light Processing (DLP). The MNs’ mechanical properties were affected by the curing temperature and duration. The shape of the MN designs and the height to aspect ratio were found to be critically important for the piercing capacity, penetration depth and penetration width of the arrays. The biological evaluation of the photocurable resin showed excellent biocompatibility in HDFs. Such mechanical integrity and biocompatibility may introduce a new field of research on the 3D printing of solid and, particularly, hollow MNs using microfluidic resin. Overall, DLP was proven to be an ideal technology for the future design and optimization of 3D-printed MNs.

## Figures and Tables

**Figure 1 micromachines-13-01368-f001:**
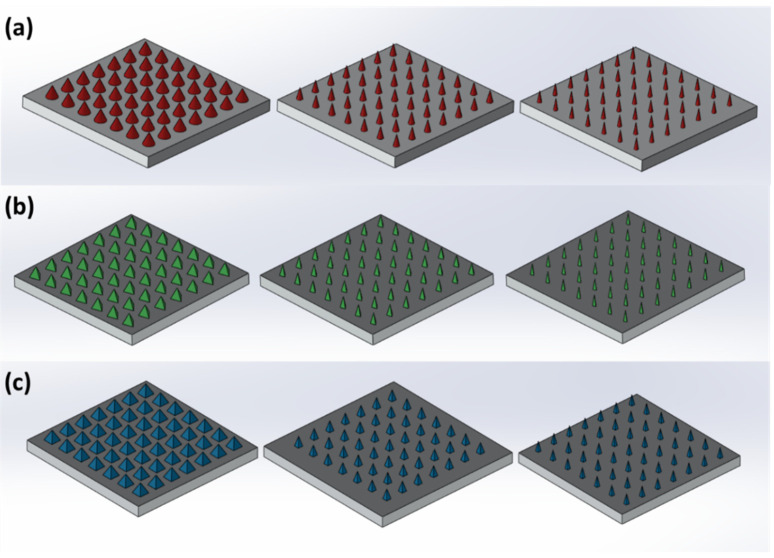
CAD design of the MNs on SolidWorks. (**a**) Pyramid, (**b**) 3-sided pyramided and (**c**) 4-sided pyramids. Height to aspect ratio from left to right (1:1, 2:1 and 3:1).

**Figure 2 micromachines-13-01368-f002:**
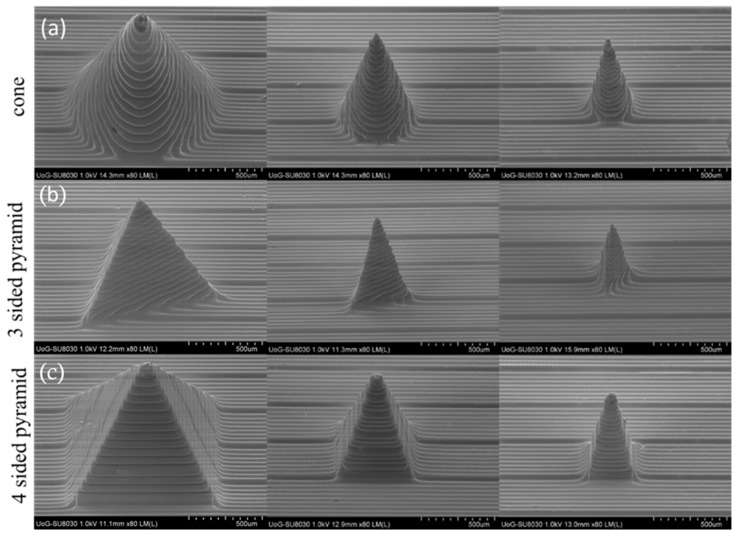
SEM images of 3D-printed MNs. (**a**) Cone, (**b**) 3-sided pyramid and (**c**) 4-sided pyramid MNs.

**Figure 3 micromachines-13-01368-f003:**
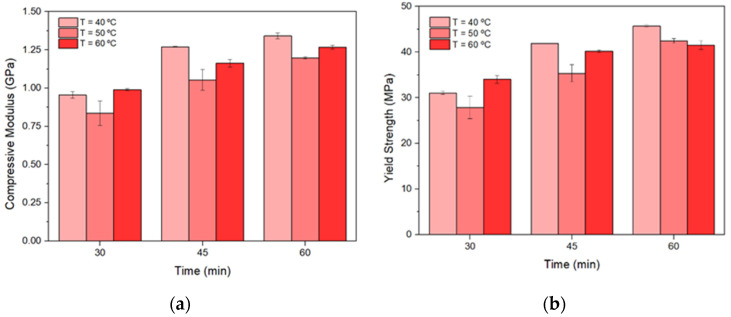
Obtained values on (**a**) compressive modulus and (**b**) yield strength by exposing all the printed cylinders to different curing settings.

**Figure 4 micromachines-13-01368-f004:**
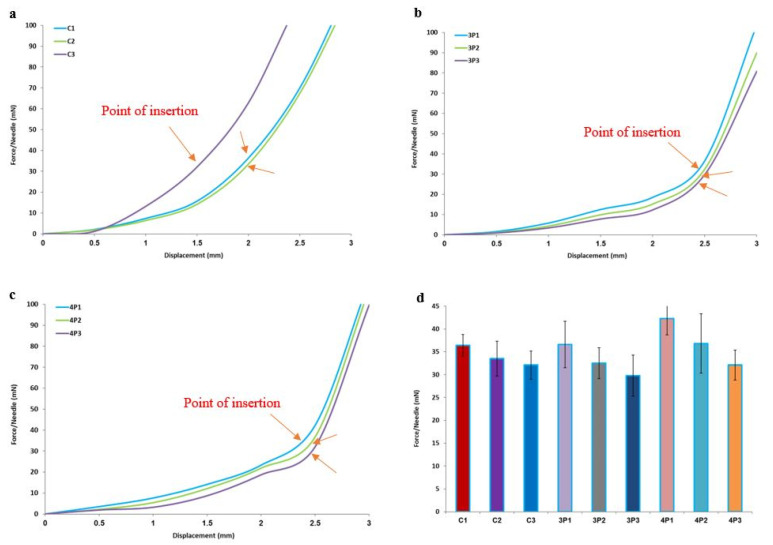
Piercing tests in porcine skin: (**a**–**c**) force/needle vs. displacement data for all designs; (**d**) insertion force/needle for all designs.

**Figure 5 micromachines-13-01368-f005:**
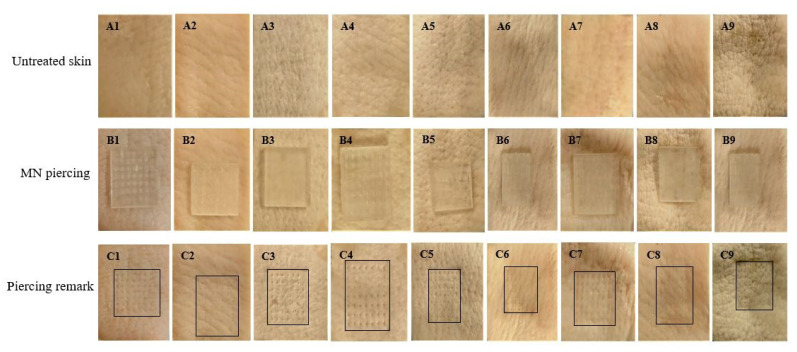
Digital photographs showing (**a**) untreated skin: A1–A9; (**b**) MN piercing: B1–B9; (**c**) piercing remarks: C1–C9 for all MN designs.

**Figure 6 micromachines-13-01368-f006:**
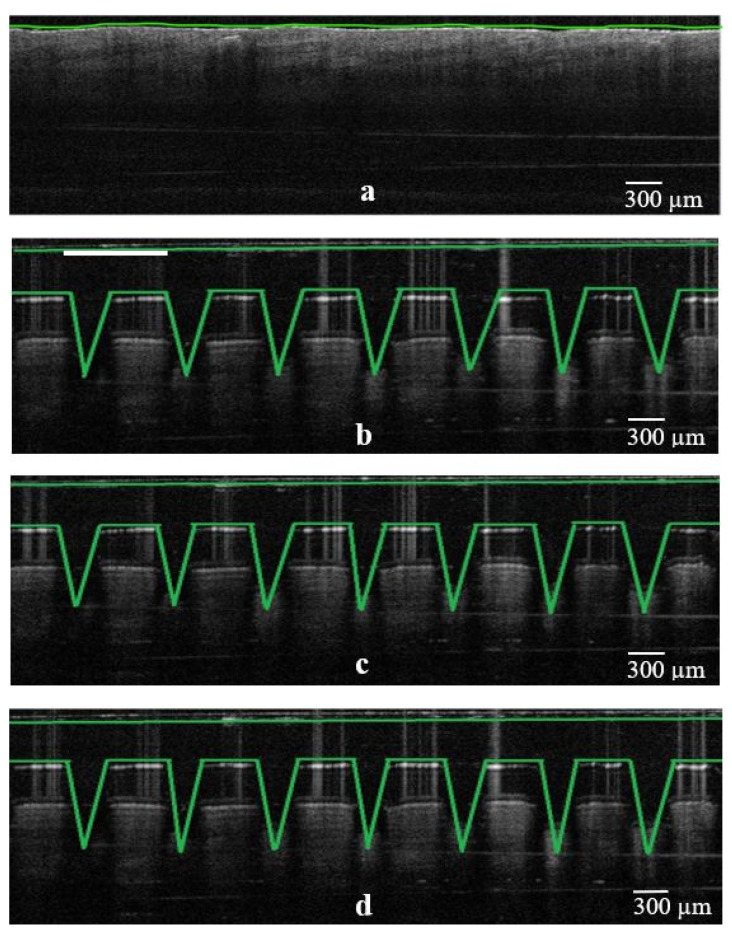
False color 2D still images of 3D-printed MNs (cone design; C1, C2 and C3) showing the real-time penetration into the mouse skin using OCT. (**a**) Skin sample as control before experiment; (**b**) insertion of C1 design; (**c**) insertion of C2 design; (**d**) insertion of C3 design, respectively (scale bar 300 µm).

**Figure 7 micromachines-13-01368-f007:**
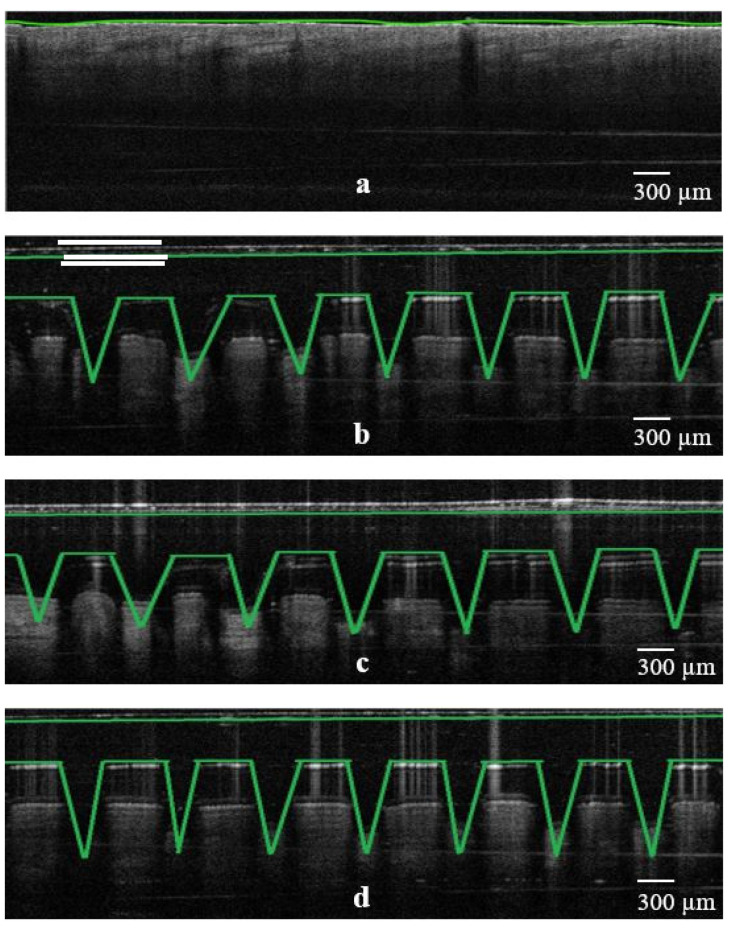
False color 2D still images of 3D-printed MNs (3-sided pyramid design; 3P1, 3P2 and 3P3) showing the real-time penetration into the mouse skin using OCT. (**a**) Skin sample as control before experiment; (**b**) insertion of 3P1 design; (**c**) insertion of 3P2 design; (**d**) insertion of 3P3 design, respectively (scale bar 300 µm).

**Figure 8 micromachines-13-01368-f008:**
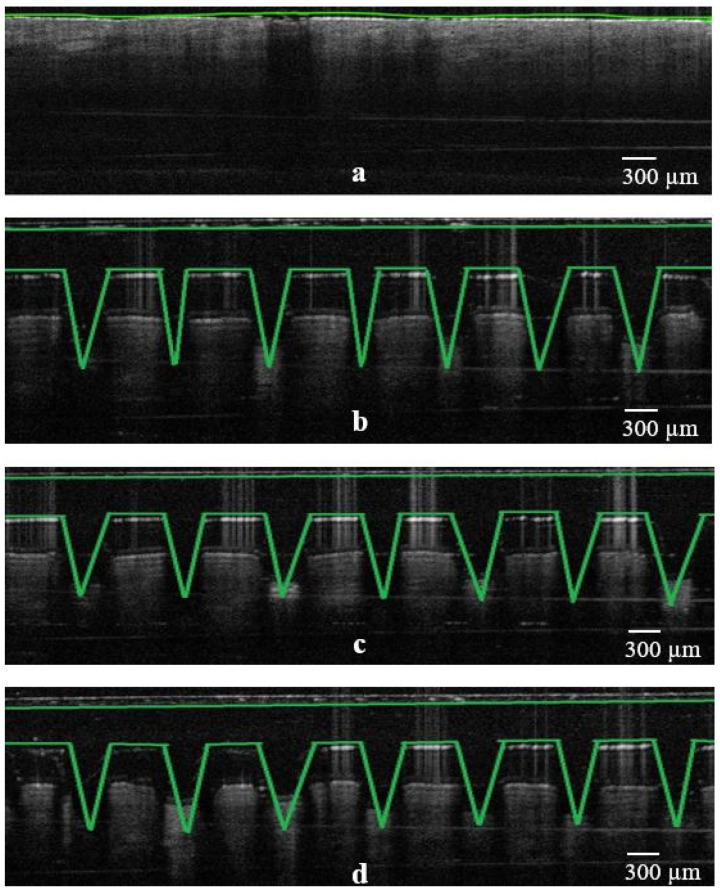
False color 2D still images of 3D-printed MNs (4-sided pyramid design; 4P1, 4P2 and 4P3) showing the real-time penetration into the mouse skin using OCT. (**a**) Skin sample as control before experiment; (**b**) insertion of 4P1 design; (**c**) insertion of 4P2 design; (**d**) insertion of 4P3 design, respectively (scale bar 100 µm).

**Figure 9 micromachines-13-01368-f009:**
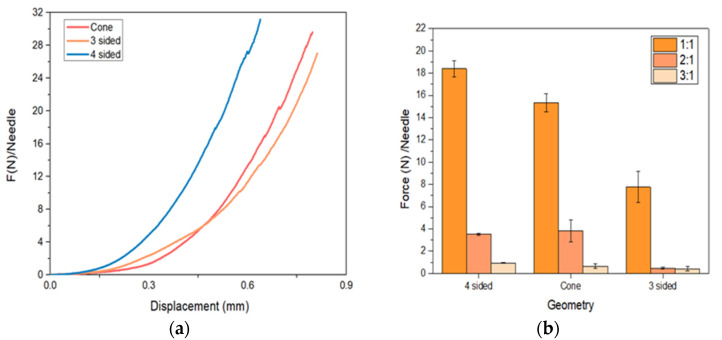
Obtained values on MN axial force measurements considering (**a**) 1:1 ratio; (**b**) different ratios and geometries.

**Figure 10 micromachines-13-01368-f010:**
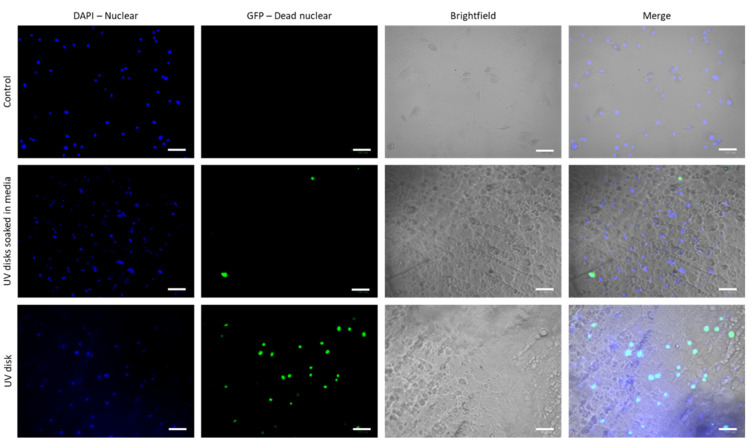
Cell imaging with blue and green staining. Human dermal fibroblasts in the concentration of 1 × 10^5^ cells per well were seeded on 96-well plates without disks (control), UV-sterilized and media-soaked disks for 2 h and 48 h, respectively, and 2 h UV-sterilized disks only. Staining was carried out according to the manufacturer’s guidelines for 30 min and plates were centrifuged before imaging via ZOE Fluorescent Cell Imager (Bio-Rad Laboratories) at 4× magnification. DAPI (blue dye) represents the nuclei of all the cells and GFP (green dye) represents the nuclei of damaged cells. The colocalization was analyzed from a minimum of 3 representative images from three independent experiments after 2.5 h incubation. Data are represented individually and as average ± SEM. Scale bar 100 µm.

**Figure 11 micromachines-13-01368-f011:**
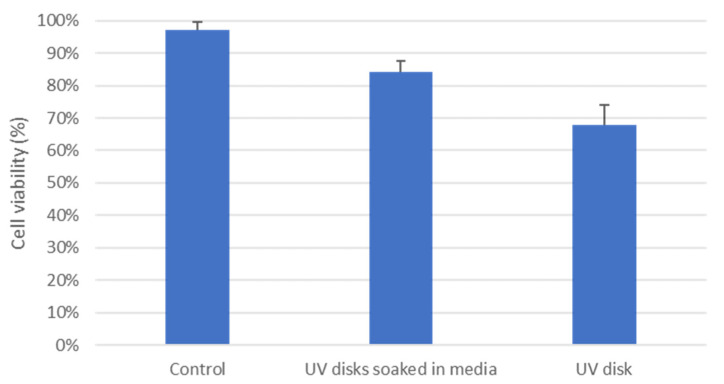
Cell viability evaluation for control, UV disks soaked in media and UV disks.

**Table 1 micromachines-13-01368-t001:** MN dimensions from CAD files.

MN Type	Height (μm)	Base (μm)
Cone 1:1 (C1)	1000	1000
Cone 2:1 (C2)	1000	500
Cone 3:1 (C3)	1000	333
3-sided pyramid 1:1 (3P1)	1000	1000
3-sided pyramid 2:1 (3P2)	1000	500
3-sided pyramid 3:1 (3P3)	1000	333
4-sided pyramid 1:1 (4P1)	1000	1000
4-sided pyramid 1:1 (4P2)	1000	500
4-sided pyramid 1:1 (4P3)	1000	333

**Table 2 micromachines-13-01368-t002:** Penetration characteristics (depth and width) in mouse skin using finger pressure (*n* = 3). C: cone, 3P: 3-sided pyramid, 4P: 4-sided pyramid.

Design	Penetration Depth (μm)	Penetration Depth (%)	Penetration Width (µm)	Penetration Width (%)
Control	0.0	0.0	0.0	0.0
C1	760.4 ± 33.2	76.0	770.2 ± 45.0	77.0
C2	801.5 ± 20.3	80.1	392.1 ± 31.0	78.4
C3	833.2 ± 24.8	83.3	298.6 ± 19.2	89.6
3P1	732.1 ± 34.1	73.2	792.1 ± 30.0	79.2
3P2	765.1 ± 36.1	76.5	404.2 ± 18.8	80.8
3P3	796.1 ± 17.3	79.6	304.4 ± 19.4	91.2
4P1	680.1 ± 33.9	68.0	809.5 ± 23.7	80.9
4P2	714.9 ± 35.1	71.4	420.5 ± 17.6	84.1
4P3	755.8 ± 26.3	75.5	309.3 ± 13.2	92.8

**Table 3 micromachines-13-01368-t003:** Obtained values for MNs’ margin of safety.

Geometry	Cone			3-Sided Pyramid			4-Sided Pyramid		
Ratio	1:1	2:1	3:1	1:1	2:1	3:1	1:1	2:1	3:1
Margin of Safety	426.2	112.7	19.8	209.6	14.6	14.0	427.6	95.0	28.5

## Data Availability

Not applicable.

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
