# Peer review of "Evaluation of 3D Printability and Biocompatibility of Microfluidic Resin for Fabrication of Solid Microneedles"

_micromachines, 2022, doi:10.3390/mi13091368_

Round 1

Author Response

We would like to thank the reviewer for the constructive comments.  We have now addressed all the queries.

This paper employed Digital Light Processing (DLP) printing technology to fabricate solid microneedle arrays. The authors explored the design of microneedle shape and height to aspect ratio influence on the arrays' piercing capacity, penetration depth, and penetration width. And the biological evaluation of the photocurable resin showed excellent biocompatibility in HDFs. The whole manuscript was well written. In view of the article's shortcomings, I recommend the publication of this manuscript after addressing the following minor comments.

  • In “1. Introduction, line 34”, “therefore such rapid prototyping approach allows the fabrication of complex and precise geometries promptly from computer-aided design (CAD) models [2].” there is no capital letter in this sentence. There were other places the capital letter was not used properly.

This has now been addressed throughout the manuscript

  • In “2. Materials and Methods”, to make this section more coherent, the positions of “2.6. Preparation of porcine skin samples “and “2.7. Microneedles axial force mechanical testing” can be exchanged.

As requested, we have exchanged the positions for respective sections

  • The equipment used is not well introduced.

We have revised the methods sections and we believe the method section is conclusive.

  • In lines 242-249, “The Materials and Methods……and appropriately cited” did not understand the meaning of the expression in the article and suggested the authors check carefully and add the intention to be expressed.

This was an error made by the authors and have been removed

  • The authors used past tense in lines 252-257, but 257-260 were written in the present tense. There were many other places the tenses were not appropriately used. This manuscript needed a major English revision.

The manuscript has now been revised and several changes have been made to meet the English standards.

  • In “3. Results”, chapters were not properly arranged, for example, 3.3. and 3.4. were missing and 3.7. was repeated.

This has now been addressed.

7)      In “3.2. Optimization of Post-Printing Curing Conditions for Enhanced Mechanical Properties”, whether the value of the ultimate strength of the specimens and the yield strength of the material were satisfied was evaluated by relevant formulas. If so, the relevant formulas should be supplemented.

We have provided the required formulas

8)      In “Table 2”, to better illustrate the strong penetration capacity of microneedles in mice skin, the authors can increase the penetration capacity of this work compared with solid microneedle arrays prepared by different printing techniques reported by others.

This is a very good comment. We have added the relevant discussion and references.

Reviewer 2 Report

The research presented in this manuscript has been well conducted with very apt testing and scientifically sound results and discussion. I don't have any further comments on potential improvement of the manuscript, but would recommend it for publication as is.

Author Response

We would like to thank the reviewer for his kind words.

Reviewer 3 Report

In this paper, the authors present a study of 3D printed microneedles (MN) and their suitability for transdermal applications. They fabricate MNs of different shapes and aspect ratios and perform a systematic investigation of the effect of different process parameters (curing time and temperature), and geometric parameters such as shape and height aspect ratio. They conclude that cone shaped MN have higher penetration depth with less applied forces. They also study the biocompatibility of the material and show that 88% of the cells survive when cultured on this surface. 

This study evaluates the use of DLP based microneedles for biological applications and how to tune the design parameters to get optimal performance. The authors have done a good job presenting a detailed explanation of all their experiments, and their reasoning is easy to follow. I recommend the publication of this manuscript, but it could use some revision to improve the quality of the discussion.

Here are some suggestions and comments - 

  • Section 2.10 - Perhaps these lines should be deleted? Page 6, Lines 242 onwards - "The Materials and Methods should be described with sufficient details to allow others to replicate and build on the published results. Please note that the publication of your manuscript implicates that you must make all materials, data, computer code, and protocols associated with the publication available to readers. Please disclose at the submission stage any re-strictions on the availability of materials or information. New methods and protocols should be described in detail while well-established methods can be briefly described and appropriately cited."

  • At several places, abbreviations are used without explanation. For instance - 

    • OCT

    • UV Disk (Fig 10)

    • In Fig. 4 explain what C1/c2/c3, 3p1 etc mean - the explanation happens at a later point in Table 2.

  • Can the authors clarify/rephrase this line -  Line 323, Page 9 "However, the slope of piercing site was constantly changing before insertion of MN tips due to continuous piercing of the skin until it reaches to a sharp drop of load"

  • Fig 6-8 - the scale bar is not visible

  • Table 2 - how is the penetration depth percentage calculated? Is it calculated based on the height of the MN?

  • Page 16, Line 438 - Please provide citation - "MNs developed by Demir et al., with forces of fracture of 0.1 N/needle and 0.5 N/needle respectively, 40 times lower than the value obtained"

  • Page 17, Line 454 - "It is obvious that by decreasing the HAS makes the MNs weaker and with a lower index" - From Table 3, decreasing the HAR from 3:1 to 1:1  increases the margin of safety. Can the authors clarify this line?

  • Section 3.8 Cytotoxicity - the data shows that the resin is cytotoxic to some extent due to the less number of living cells, and the authors state that the resin is suitable for transdermal applications. The cells on the disks are less viable than the control - does this mean that the UV disk has some cytotoxicity? Even in Figure 11, the soaked disks have more green (damaged) cells than the control. Can the authors comment on this - is this an acceptable level of cytotoxicity given that there's a 12% drop in living cells?

  • The scale bar is not visible in Fig. 11

Author Response

We would like to thank the reviewer for the constructive comments. We have now addressed all the queries.

  • Section 2.10 - Perhaps these lines should be deleted? Page 6, Lines 242 onwards - "The Materials and Methods should be described with sufficient details to allow others to replicate and build on the published results. Please note that the publication of your manuscript implicates that you must make all materials, data, computer code, and protocols associated with the publication available to readers. Please disclose at the submission stage any re-strictions on the availability of materials or information. New methods and protocols should be described in detail while well-established methods can be briefly described and appropriately cited."

This has now been addressed

  • At several places, abbreviations are used without explanation. For instance - 
  •  
    • OCT

Full term was used for OCT when used the first time in materials and methods section

    • UV Disk (Fig 10)

Full term for the abbreviation is now provided for the first time it was used in the manuscript

    • In Fig. 4 explain what C1/c2/c3, 3p1 etc mean - the explanation happens at a later point in Table 2.

This how now been addressed in Table1 along with the MN dimensions

  • Can the authors clarify/rephrase this line -  Line 323, Page 9 "However, the slope of piercing site was constantly changing before insertion of MN tips due to continuous piercing of the skin until it reaches to a sharp drop of load"

The sentence has now been rephrased

  • Fig 6-8 - the scale bar is not visible

The  images have been revised and the scale bars can be clearly seen.

  • Table 2 - how is the penetration depth percentage calculated? Is it calculated based on the height of the MN?

Yes, it has been calculated based on the total length.  Additional discussion has been added for comparison with dissolvable polymeric microneedles

  • Page 16, Line 438 - Please provide citation - "MNs developed by Demir et al., with forces of fracture of 0.1 N/needle and 0.5 N/needle respectively, 40 times lower than the value obtained"
  • This is now added

  • Page 17, Line 454 - "It is obvious that by decreasing the HAS makes the MNs weaker and with a lower index" - From Table 3, decreasing the HAR from 3:1 to 1:1 increases the margin of safety. Can the authors clarify this line?

This has been clarified and addressed 

  • Section 3.8 Cytotoxicity - the data shows that the resin is cytotoxic to some extent due to the less number of living cells, and the authors state that the resin is suitable for transdermal applications. The cells on the disks are less viable than the control - does this mean that the UV disk has some cytotoxicity? Even in Figure 11, the soaked disks have more green (damaged) cells than the control. Can the authors comment on this - is this an acceptable level of cytotoxicity given that there's a 12% drop in living cells?

We have performed the experiments again to have more representative data and much more comparable microscopic images.   In a normal cell culture, you expect to have a viability of 90 to 100%. Therefore, having a cell viability of above 80 % for the disks is very good which indicates the resin toxicity is at its minimum. 

  • The scale bar is not visible in Fig. 11

The scale bars are now visible,

Round 2

Reviewer 1 Report

The whole manuscript was well written. The questions are well answered.